# Optimizing heat pump unit selection for recirculating aquaculture workshops through computational fluid dynamics

**Xu Ziyun**[1], **Du Ping**[1,2]\*, **Wan Jiacheng**[1], **Lin Leyao**[3], **Chen Kairui**[1]

**1** College of Mechanical and Power Engineering, Dalian Ocean University, Dalian, Liaoning, China, **2** Key Laboratory of Environment Controlled Aquaculture (Dalian Ocean University) Ministry of Education, Dalian, Liaoning, China, **3** College of Locomotive and Vehicle Engineering, Dalian University of Transport, Dalian, Liaoning, China

\* xu.ziyun00qq852741@hotmail.com

**Data Availability Statement:** All relevant data are within the manuscript and its Supporting Information files.

## Abstract

The selection of water temperature regulation equipment plays a crucial role in the design of workshops. At present, the choice of water temperature control equipment is usually based on the volume of the fish pond and thermal parameter calculation, combined with aquaculture experience. Empirical formulas only work in specific conditions due to factors like the environment, climate, and fish types,resulting in inaccurate equipment selection outcomes. Recognizing this limitation, this paper proposes to apply CFD simulation of the temperature field to accurately calculate the heat exchange value between indoor air and water, thereby predicting the heat exchange values during aquaculture activities in the aquaculture workshop. providing a new approach for equipment selection. This paper selects a puffer fish breeding workshop in Dalian as the simulation object, establishing a 3D unsteady-state Computational Fluid Dynamics model. The model considers outdoor temperature, solar radiation, and phase-change heat transfer in water. Comparison with experimental data reveals a root mean square error of 0.46°C for the simulated results. During summer, the highest cooling load occurs at 16:00, reaching 94.6 kW. It is recommended to employ the Daikin GCHP-40MAH ground source heat pump as the water temperature control equipment. CFD simulation validates its effectiveness in shaping the indoor temperature field post-installation. the investment in water temperature control equipment can be reduced to a certain degree. This provides a reference value for the selection of water temperature equipment in aquaculture workshops.

## 1. Introduction

In recent years, with the rapid development and expansion of the aquaculture industry, factory-based recirculating aquaculture workshops have become one of the main forms of aquaculture, receiving widespread application. Compared to traditional aquaculture, factory-based recirculating water aquaculture has the advantages of saving land area, water, high-density aquaculture, and easy control of water temperature. It aligns with the requirements for sustainable development in China and is the future trend of transformation in aquaculture methods [1].

**Funding:** The author(s) received no specific funding for this work.

**Competing interests:** The authors have declared that no competing interests exist.

CFD, an abbreviation for Computational Fluid Dynamics, is a technology that uses numerical methods to simulate fluid motion, heat, and mass transfer processes. Compared to the energy consumption calculations for traditional water temperature control equipment selection, CFD simulation can establish a more detailed mathematical model based on the actual situation, taking into account various factors that affect the temperature field, such as solar radiation, water temperature, and workshop location, and outdoor temperature changes over time, thereby improving the calculation accuracy. CFD numerical simulation technology has been widely used for related calculations in the field of agricultural engineering. For example, Villagrán and colleagues [2] utilized a two-dimensional transient Computational Fluid Dynamics (CFD) model to simulate the indoor airflow field in the high-latitude tropical region of the Andes. This model successfully predicted the air temperature and airflow distribution within the DMG greenhouse. Bartzanas and colleagues [3] employed a three-dimensional CFD computational model to simulate the ventilation conditions of a tunnel greenhouse equipped with insect-proof nets.

CFD plays a crucial role in accurately selecting heat pumps for recirculating aquaculture systems, providing precise heat load calculations tailored to diverse building characteristics and geographical environments. This significantly reduces the investment in water temperature control equipment.ANSYS is a finite element analysis software developed by the American company ANSYS in 1970, integrating capabilities for structural, fluid, electrical, and magnetic field analysis. Widely employed across various industries, it is particularly prevalent in simulating temperature fields in greenhouse environments for vegetable cultivation and has gradually found applications in temperature field simulations for aquaculture workshops in recent years. In summary, this paper utilizes Ansys Fluent as the tool for simulating the temperature field within the workshop, aiming to predict the cooling load and facilitate equipment selection.

He and colleagues [4] conducted a study using a two-dimensional Computational Fluid Dynamics transient model to investigate the influence of the size of the back wall ventilation port on the temperature and airflow distribution within a solar greenhouse. In a separate study, Nebbali and colleagues [5] employed a three-dimensional CFD transient model to numerically simulate the distribution of climate parameters within a tomato greenhouse. Their research explored the comprehensive effects of the sun's position, wind direction, and intensity on the microclimate within the greenhouse. Shenghan Zhou and colleagues [6] conducted a study using FLUENT software to simulate the temperature field in a plant factory. Their research focused on the detailed three-dimensional modeling of temperature distribution influenced by LED light heat dissipation and boundary conditions, emphasizing the significance of accurate modeling in such environments. Long Yang and his team [7] utilized FLUENT in their research to simulate the air flow and temperature fields in a gosling house with vertical wall attached jet ventilation in a cold region. This study investigated the effects of different air supply velocities on the thermal environment, providing essential insights for managing ventilation in livestock housing under cold climate conditions. Cruz Ernesto Aguilar Rodriguez and collaborators [8] applied FLUENT software in their research for simulating heat and mass transfer processes in greenhouses. Their work highlighted the importance of heating systems like pipe heating and its impact on crop yields, energy consumption, and operational costs. The study employed computational fluid dynamics to analyze factors such as air movement, temperature gradients, and crop transpiration, crucial for understanding the dynamics of greenhouse systems. The aforementioned studies demonstrate that CFD has been widely applied in the agricultural field, and the use of Fluent software for temperature field simulations is quite mature.

In this study, CFD technology is used to predict the heat exchange between the water body and the indoor environment, enabling precise calculation of the cooling load to select temperature control equipment.

For such breeding workshops, the control requirements for indoor air temperature are relatively low, and they are usually not equipped with air conditioning equipment for active temperature regulation. Therefore, the actual value of air temperature is largely determined by the temperature changes in the external environment and the initial temperature distribution, making it difficult to effectively predict the temperature field distribution within the workshop. There has been some research on the impact of building wall heat storage on the temperature field. Tiwari and others [9] compared different heating technologies in non-air-conditioned buildings and found that water wall heat storage can reduce indoor temperature fluctuations by increasing specific heat. Moustafa and others [10] used transient CFD numerical simulation to evaluate pottery water walls under arid and hot climates, exploring the cooling and heating efficiency of pottery water walls. The paper shows that the transient CFD model can simulate the impact of heat storage walls on the indoor temperature field over time. Sodha and others [11] analyzed the impact of heat storage on the thermal performance of non-air-conditioned rooms. The results showed that heat storage materials reduced the fluctuations in the indoor temperature field, and both the thermal conductivity of the heat storage material and the specific heat capacity of the heat storage material have an impact on temperature changes.

In factory-based recirculating aquaculture workshops, the selection of water temperature control equipment directly affects the benefits and production costs of aquaculture [1]. Therefore, the selection of water temperature regulation equipment plays a crucial role in the workshop design scheme. At present, the common method of equipment selection is to calculate the volume of the fish pond and select the appropriate water temperature control equipment combined with the heat capacity, thermal inertia, and other parameters of the water and breeding experience. Since empirical formulas are usually derived based on empirical data and assumed environmental conditions, their application range is relatively limited. Under different environmental conditions, climate characteristics, fish species, and growth stage conditions, the cooling load values are different, and the empirical formula lacks comprehensive applicability. To ensure the accuracy and feasibility of the selected water temperature control equipment, in addition to calculating the above parameters, it is also necessary to consider factors such as the area and height of the workshop, local solar radiation intensity, and phase changes on the water surface. Therefore, in the equipment selection process, these factors need to be considered comprehensively to develop a reasonable calculation scheme and obtain more accurate equipment selection results.

## 2. Establishment and solution of the CFD model for factory-based recirculating aquaculture workshops

### 2.1 Physical model of the workshop space

The aquaculture workshop is located in Dalian, Liaoning Province, with geographic coordinates of 122˚29' E longitude and 39˚36' N latitude.The workshop's architectural schematic is as shown in Fig 1. The workshop is constructed with a steel structure, with the ridge running from east to west, a width of 20m, shoulder height of 3.6m, eave height of 4.4m, and a length of 61.75m. The workshop roof is equipped with 10 semi-transparent resin light strips, each 0.47m wide and 6.9m long, with a light transmittance of 0.8, to introduce natural lighting. The workshop is equipped with 10 octagonal aquaculture ponds, each with an area of 53.26m$^2$ and a depth of 2.5m, and a 1.5m aisle space is set between adjacent aquaculture ponds.The architectural frontal and overhead views of the workshop are illustrated in Figs 2 and 3, respectively, as presented in this section.

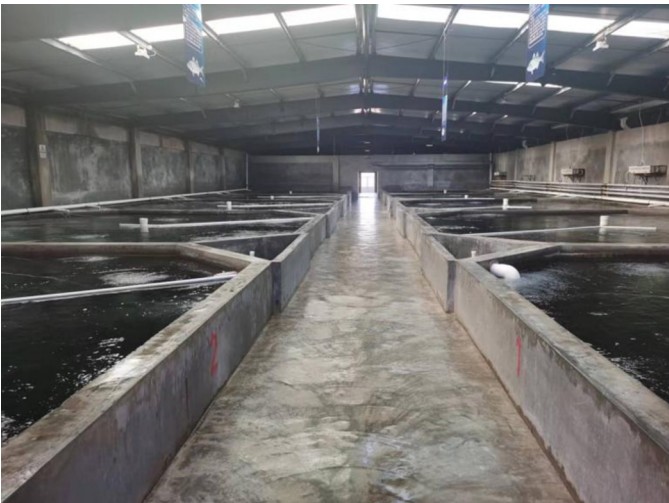

**Fig 1. Architectural diagram of the recirculating aquaculture workshop.**

## 2.2 Computational domain and grid division

This study analyzed the air energy exchange in the factory-style circulating water breeding workshop, identifying the main parts that exchange heat with the indoor air, including building walls, water bodies, and pool walls. We used a fluid-solid coupling simulation method to describe the energy transfer process indoors and accordingly divided the entire building workshop model into four computational domains: building enclosure walls, indoor air, pools, and water bodies [12]. The grid division setting needs to consider the data transfer method of the computational domain. Data transfer exists between the contact surfaces of different computational domains. The contact surfaces between various computational domains in the model are set as shared topological surfaces, and the grid of the two contact surfaces is kept consistent. The order of grid generation in the computational domain of this model determines the

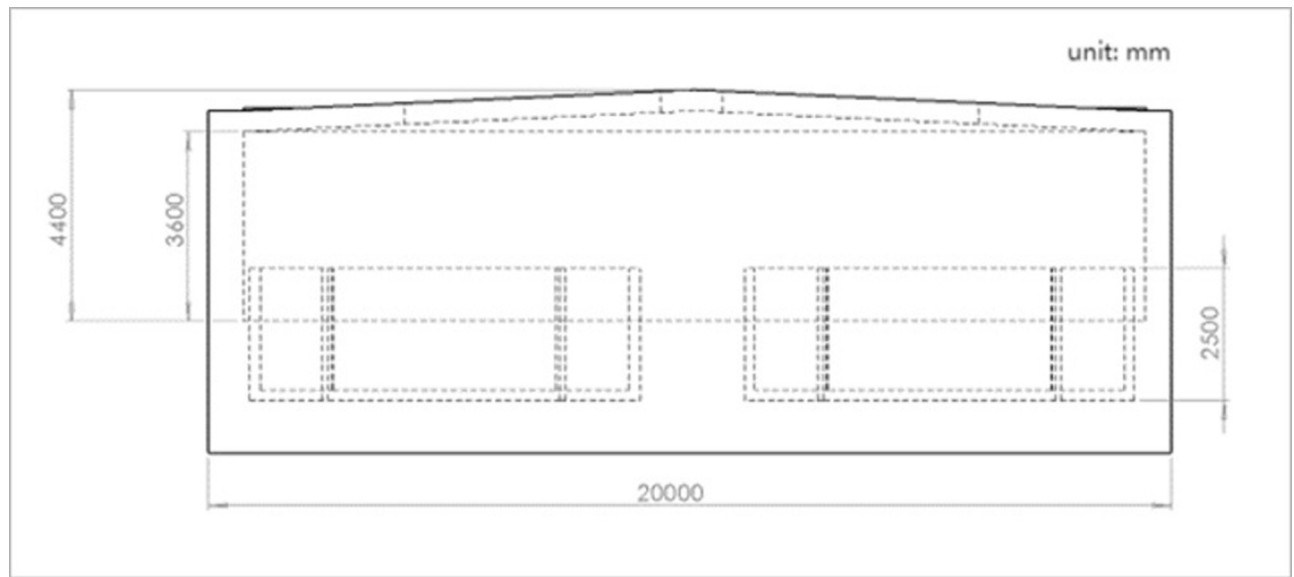

**Fig 2. Main view of the recirculating aquaculture workshop.**

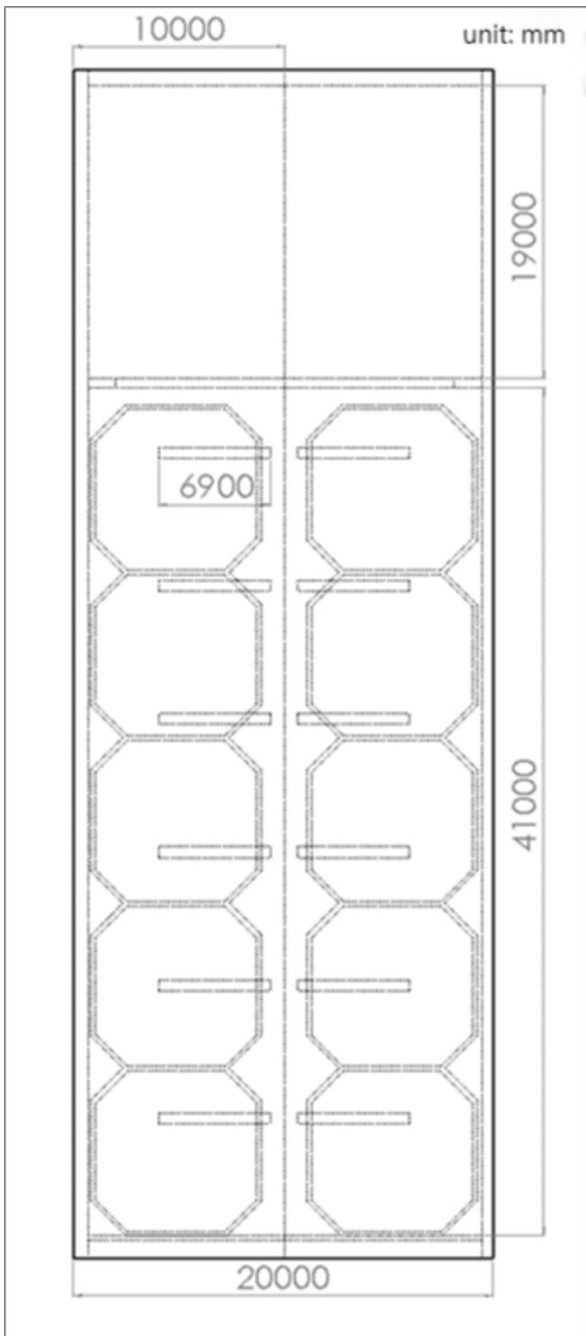

**Fig 3. Top view of the recirculating aquaculture workshop.**

overall structure of the grid. To ensure calculation accuracy, the order of dividing the model grid is as follows: aquaculture water body, pool, air, and building enclosure.

This paper designs three different schemes for the grid division size of each computational domain in the workshop, as shown in Table 1.

Among them, the results of the computational domain grid division in Scheme 2 are shown in Fig 4. This paper monitors the change in the highest temperature of the indoor air under

**Table 1. Grid division schemes of grid independence experiments.**

|  | Computational Domain | Grid Size (m) | Number of Grids |
|---|---|---|---|
| Scheme 1 | Water Body | 0.3 | 378100 |
|  | Pool | 0.3 | 94525 |
|  | Air | 0.4 | 567150 |
|  | Building Enclosure | 0.3 | 189050 |
| Scheme 2 | Water Body | 0.2 | 1417857 |
|  | Pool | 0.2 | 1063393 |
|  | Air | 0.3 | 2126786 |
|  | Building Enclosure | 0.2 | 354464 |
| Scheme 3 | Water Body | 0.1 | 6180441 |
|  | Pool | 0.1 | 1545110 |
|  | Air | 0.2 | 9270662 |
|  | Building Enclosure | 0.1 | 4635331 |

the meteorological parameter conditions on October 20, 2022. The highest temperature of the indoor air change with grid refinement as shown in the following Fig 5.When the grid division adopts Scheme 2, the highest indoor temperature tends to stabilize. Therefore, this paper adopts the grid division method shown in Scheme 2.

## 2.3 Basic control equations

When conducting numerical simulations, Computational Fluid Dynamics (CFD) follows three major conservation laws. These laws form the basis of the equations that Fluent software

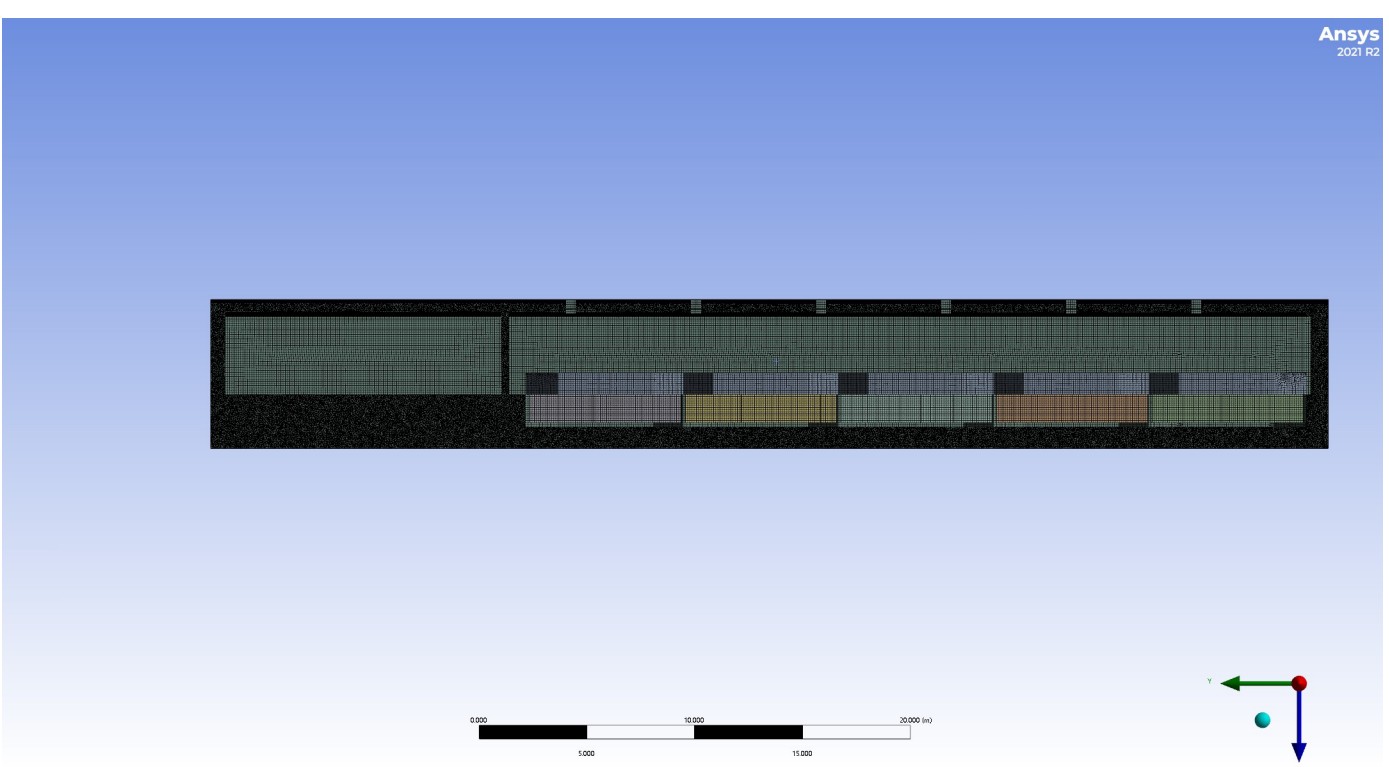

**Fig 4. Schematic diagram of the grid division for the computational domains of the second scheme.**

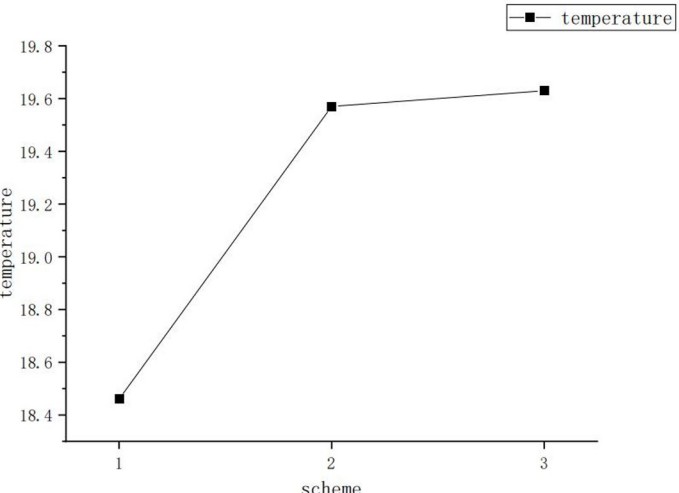

**Fig 5. Grid independence line chart.**

relies on when simulating physical fields. When considering the air flow and heat transfer process in the workshop, we follow the conservation equations of mass, momentum, and heat, the general forms of which are [13]:

$$\frac{\partial(\rho\varphi)}{\partial t} + \mathrm{div}\left(\rho\vec{v}\,\varphi + J_\varphi\right) = S_\varphi \tag{1}$$

$$J_\varphi = -\Gamma_\varphi \mathrm{grad}\varphi \tag{2}$$

In the formula:

$\varphi$- the general parameter variable;

$t$- time;

$\vec{v}$ - the vector of air flow speed, m/s;

$J_\varphi$- the diffusion flux;

$S_\varphi$- the generalized source term;

$\Gamma_\varphi$- the generalized diffusion coefficient, m$^2$/s.

When $\varphi = 1$ in the equation, it is the continuity equation; when $\varphi = \vec{v} = [\vec{u}, \vec{v}, \vec{w}]$, it is the momentum equation; and when $\varphi = T$, it is the energy equation. When phenomena such as phase change, radiation, and convection occur, causing temperature changes, the air density changes accordingly, which then affects the momentum equation and influences air flow, realizing multi-model coupling.

The aquaculture workshop, as a closed system, only exchanges heat with the outside world and does not exchange mass with the outside world. The state of indoor air can be regarded as three-dimensional, non-steady-state, incompressible Newtonian fluid motion. Due to the temperature difference between the water body and the walls in the workshop, which leads to an inconsistent distribution of air density indoors, natural convection of indoor air is easily induced in the workshop. The fluid flow is slow, as measured by a hot-wire anemometer, indicating a near-wall air velocity of 0.01 m/s. Using the height of the workshop as the characteristic length, the calculated Reynolds number is 2000, which falls within the laminar flow regime. so the laminar flow model is used for simulation calculation [14, 15]. Since the temperature change in the workshop environment is not significant, within this temperature change range, the density change with

temperature can be approximated as linear, and its density change is far less than the density itself. Therefore, by considering the effects of air density and gravity due to temperature changes, and employing the Boussinesq assumption, the buoyancy induced by temperature differences is simplified and incorporated into the source term of the energy equation [16].

## 2.4 Selection of computational models

**2.4.1 Radiation model.** The workshop interior exchanges radiative heat with the external environment. The Discrete Ordinates (DO) radiation model, based on discrete coordinates, is applicable to various optical thicknesses. This model not only considers the scattering effect during radiation transfer but considers radiation direction. It can describe the specular reflection and scattering effects produced by reflective water surfaces. The semi-transparent media in the workshop are water and air, both with relatively small optical thicknesses. At the same time, open water surfaces occupy a large area inside the workshop. Considering the refraction, absorption, and reflection when radiation enters the water body, as well as the scattering of solar radiation through the light-transmitting board, this paper adopts the DO model to describe the radiative heat exchange between various surfaces in the workshop [17].

**2.4.2 Phase change mode.** The evaporation process occurring between the water surface and the air in the workshop cannot be neglected. The evaporation process not only causes mass transfer, but also exchanges latent heat, affecting the temperature field distribution [18]. In this paper, a User-Defined Function (UDF) is used to add the phase change latent heat to the heat transfer equation in the form of a source term. By scanning the air computation domain, the grids in contact with the water surface in the computation domain are determined. Based on these grids, the air stream speed above is extracted, and the water surface evaporation rate is solved using an empirical formula, the heat absorbed by the water body during phase change is calculated by considering the change in enthalpy values associated with evaporation. This heat is then incorporated into the energy equation as a source term, allowing for a more accurate simulation of temperature field variations. The calculation method refers to different references and works, considering factors such as the wind speed on the water surface and the partial pressure of water vapor, and obtains accurate results of the water surface evaporation rate and energy source term. One of the empirical formulas is particularly suitable for scenarios where the flow speed is less than 0.5 m/s, aligning well with the parameters of this study. The empirical formula for calculating the water surface phase change energy source term can be used as follows [19]:

$$S_{\text{eva}} = (\alpha + 0.00013v) \times (P_2 - P_1) \times A \times \frac{B}{B'} \tag{3}$$

$S_{\text{eva}}$- Water surface evaporation mass rate, g/s;

$\alpha$- Diffusion coefficient. This value is related to the water temperature. When the water temperature is 17°C, it is taken as 0.00017kg/(m²·h·Pa);

$v$- Air flow speed on the water surface. In the text, it refers to the air flow speed on the open water surface, m/s;

$P_2$- The partial pressure of saturated water vapor at the water temperature, 3556Pa;

$P_1$- The actual partial pressure of water vapor in the air in the workshop, 2655Pa;

$A$- The area of the pool water surface, m²;

$B$- Standard atmospheric pressure, 101325Pa;

$B'$- The actual atmospheric pressure at the location. The atmospheric pressure at the location of the workshop in winter is about 101300Pa.

The energy source term from water surface evaporation $S_{h1}$ is as follows:

$$S_{h1} = S_{\text{eva}} \cdot h_{lg} \tag{4}$$

$h_{lg}$-Latent Heat of Vaporization of Water Vapor,J/kg.

## 2.5 Computational domain and boundary conditions

The workshop model includes four-types of computational domains: building envelope, indoor air, pool, and water body. The properties of these materials are as shown in Table 2 [20].

The surfaces of the building envelope in contact with the external environment are set as third-type boundary conditions that vary with time. The surfaces of the building envelope in contact with adjacent workshops is set as a symmetric boundary conditions. The surfaces of the daylighting strip are also set as third-type boundary conditions. The surfaces of the building bottoms in contact with the soil are set as first-type boundary conditions.

(1) Ambient Temperature

In this study, temperature control equipment LogTag TRIX-8, as shown in Fig 6, was placed 15 meters southwest of the workshop, recording the outdoor temperature changes over a 24-hour period at 15-minute intervals. The collected data were fitted using a third-order Fourier's law function in MATLAB, simplifying it to a 24-hour periodic simple harmonic motion [3]. As shown in Fig 7, the outdoor temperature parameter fitting curve for October 20, 2022, has a root-mean-square error (RMSE) of 0.6302.The equation for the outdoor temperature at any time is as follows [21]:

$$
\begin{aligned}
T_0(t) = {} & 0.09096\cos(7.0t \times w) - 0.1263\cos(3.0t \times w) - \\
& 0.009609\cos(6.0t \times w) - 0.5131\sin(3.0t \times w) - \\
& 0.1213\sin(6.0t \times w) - 0.1849\sin(5.0t \times w) - \\
& 0.2411\sin(7.0t \times w) - 2.69\cos(t \times w) - 4.229\sin(t \times w) + \\
& 1.343\cos(2.0t \times w) - 0.1611\cos(4.0t \times w) - \\
& 0.001382\cos(8.0t \times w) + 0.1782\sin(2.0t \times w) - \\
& 0.2462\sin(4.0t \times w) + 0.1184\sin(8.0t \times w) + \\
& 0.0815\cos(5.0t \times w) + 11.57
\end{aligned} \tag{5}
$$

In the equation, W = 7.93*e-5.

**Table 2. Material properties setting table.**

| project | air | water | concrete | plane skylight |
|---|---|---|---|---|
| Density (kg/m$^3$) | Ideal gas Density varies with temperature | 1000 | 2100 | 1500 |
| specific heat (J/kg·k) | 1006.43 | 4200 | 880 | 1050 |
| heat conductivity coefficient (W/m·k) | 0.242 | 0.599 | 1.4 | 0.158 |
| absorptivity | 0 | 0.1 | 0.6 | 0.1 |
| scattering coefficient | 0 | 0 | 1 | 0 |
| diffusion coefficient | 1 | 1 | 1 | 1 |
| emissivity | 0.86 | 0.85 | 0.71 | 0.85 |

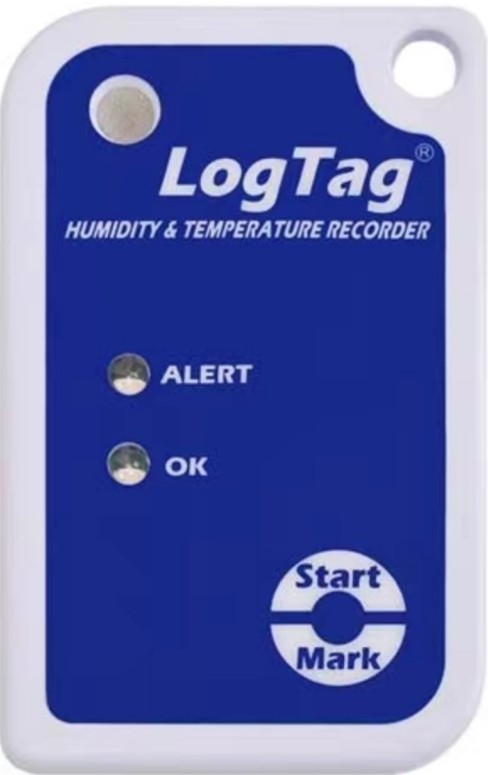

**Fig 6. LogTag TRIX-8 temperature and humidity sensor.**

(2) Solar Radiation

The solar radiation calculation module in Fluent is used to automatically compute the local solar radiation intensity, with the calculation results applied to the building exterior wall for energy transfer calculations in the form of second type boundary conditions. Based on the geographical location of the Zhuanghe area and the building orientation, the solar azimuth, direct

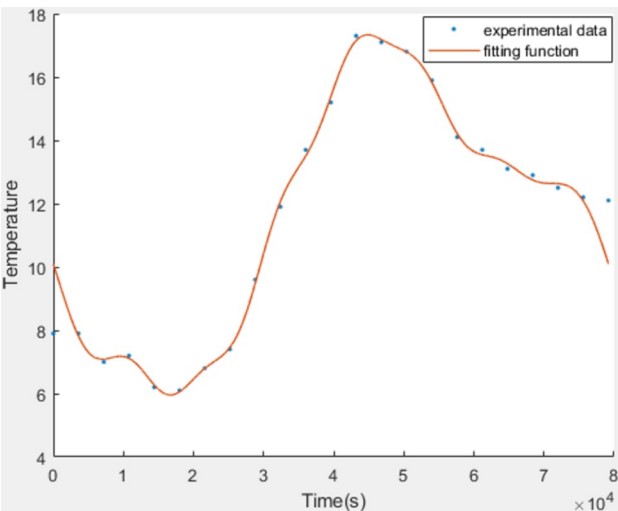

**Fig 7. Measured outdoor temperature and fitting curve.**

radiation, and scattered radiation for a specific date and location were calculated. In unsteady-state calculations, the solar radiation calculation is updated every 10 time steps to obtain the solar radiation azimuth and radiation intensity that vary with time, thereby setting the dynamic boundary conditions for outdoor solar radiation [22].

## 2.6 Initial conditions and calculation method

In the simulation of air energy transfer within the workshop, the external environment and the constant temperature of water body serve as the primary sources of energy. Taking into account the influence of these factors, the thermal environment changes within the workshop from 6:00 to 18:00 on October 20, 2022, were simulated. The finite volume method was adopted to discretize the model equations, divide the computational domain for the numerical simulation into a grid, ensuring that there is a unique control volume around each grid point. Obtain discrete equations by integrating the differential equations. The Semi-Implicit Method for Pressure-Linked Equations (SIMPLE) was utilized to solve the steady-state temperature distribution at 6:00 AM, which was then used as the initial condition. Subsequently, the Pressure Implicit with Splitting of Operators (PISO) method was selected for unsteady-state calculations. A semi-implicit format was chosen for time discretization, to reduce the discretization error caused by time discretization by obtaining the average of explicit and implicit schemes. with a time step set at 30 seconds, and the maximum time step number for each iteration was set to 100 steps.

## 3. Experimental design and model validation for a factory-based recirculating aquaculture workshop

### 3.1 Experimental design

The experiment was conducted over three days, starting from October 20th to 22nd, 2022, in a pufferfish breeding workshop in Zhuanghe, Dalian. Indoor temperature parameters were collected and automatically saved by the LogTag TRIX-8 standard temperature recorder, with a collection interval of 15 minutes. Using a combination of a cross-type and three-dimensional net structure, a central test point is established in the center of the aquaculture room. Based on this, test points are uniformly arranged along the north-south axis at intervals of 8 meters, and along the east-west axis at intervals of 9 meters, together forming a "cross" shaped layout. Vertically, in the workshop's corridors and equipment areas, three sensors are layered at heights of 0.2m, 1.5m, and 3m. Additionally, two sensors are also layered vertically above the water pool, at heights of 1.5m and 3m, respectively. In total, 39 temperature monitoring points are set up. The arrangement is shown in Fig 8.

### 3.2 Model validation

Fig 9 presents the error distribution diagram comparing the numerical calculation results at all detection points within the workshop at each moment with the actual measured data. The maximum absolute error between the simulated values and the actual data was 1.4˚C, with a root mean square error (RMSE) of 0.46˚C and an average error of 0.24˚C.

Fig 10 compares the simulated and measured values of the average temperature at three levels in the vertical direction on October 20th. The diagram shows that the indoor temperature changes lag behind the outdoor environmental temperature changes. When the outdoor temperature reaches a turning point, the indoor temperature does not immediately respond with a corresponding change, but continues to maintain its original trend. After a certain period, the change in room temperature will occur. The simulated values match the measured indoor

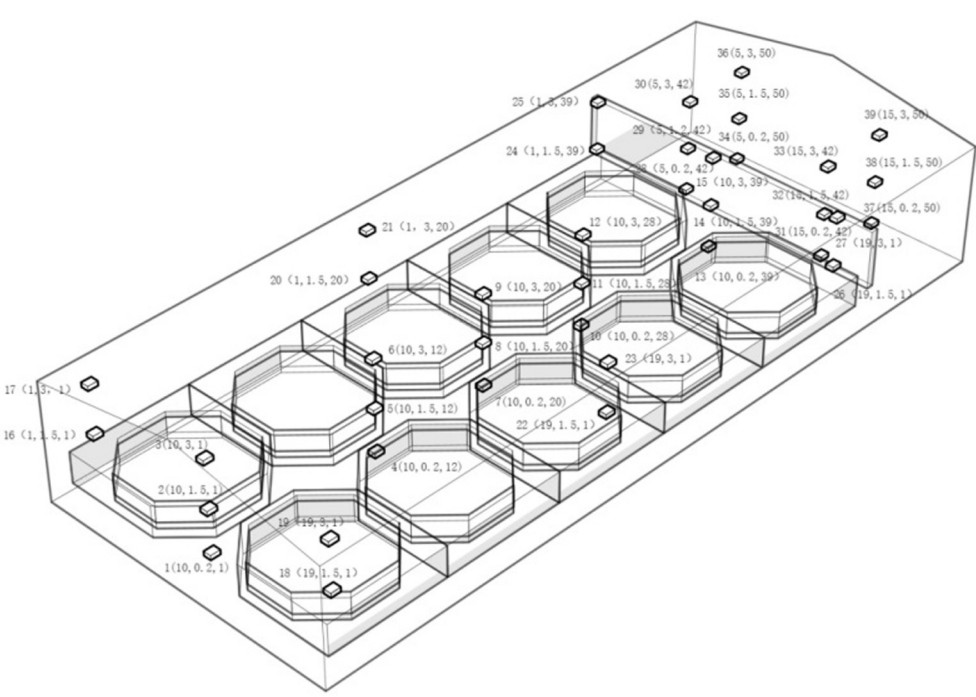

**Fig 8. Temperature measurement layout in factory aquaculture workshops.**

temperature changes during this period, indicating that this transient model can predict the indoor temperature field under different air temperature changes [23].

## 4. Equipment selection for recirculating aquaculture workshop

### 4.1 Hourly air temperature simulation inside the workshop in summer

Using the aforementioned model, we can calculate the air temperature inside the workshop during the summer. According to the "Air Conditioning Design Manual", the formula for calculating the hourly temperature of the external environment of the workshop in summer is as follows [24]:

$$t_{sh} = t_{wp} + \beta \Delta t_r \tag{6}$$

$t_{sh}$-The calculated hourly temperature outside during summer,°C;

$t_{wp}$-The calculated average daily temperature outside during summer,°C,The average daily exterior temperature of the workshop in the city of Dalian during summer is 25.5°C;

$\beta$-The hourly coefficient of the outdoor temperature, which is determined according to Tables 3 and 4.

$\Delta t_r$-The average daily temperature difference in the outdoor environment during summer should be calculated as per the following formula:

$$\Delta t_r = \frac{t_{wg} - t_{wp}}{0.52} \tag{7}$$

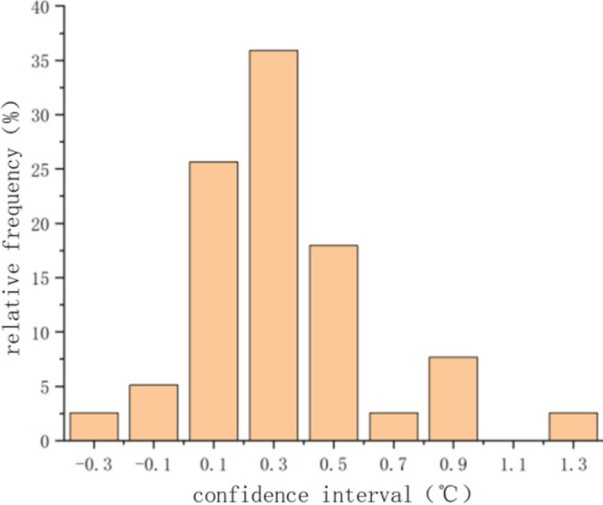

**Fig 9. Error distribution diagram between imulated and measured values.**

$t_{wg}$-The calculated dry bulb temperature outside during summer,˚C,The calculated dry bulb temperature outdoors in the city of Dalian during summer is 27.4˚C.

The hourly outdoor temperature in Dalian during the summer is calculated. The data for the hourly outdoor temperature in Dalian during the summer is then fitted with a function. This fitted function is set as the boundary condition for the numerical simulation. Using a solar radiation calculator, the initial date is set as July 23, 2022 (the height of summer). This allows for the calculation of the temperature distribution in a factory's circulating water workshop in the North under typical summer climate conditions.

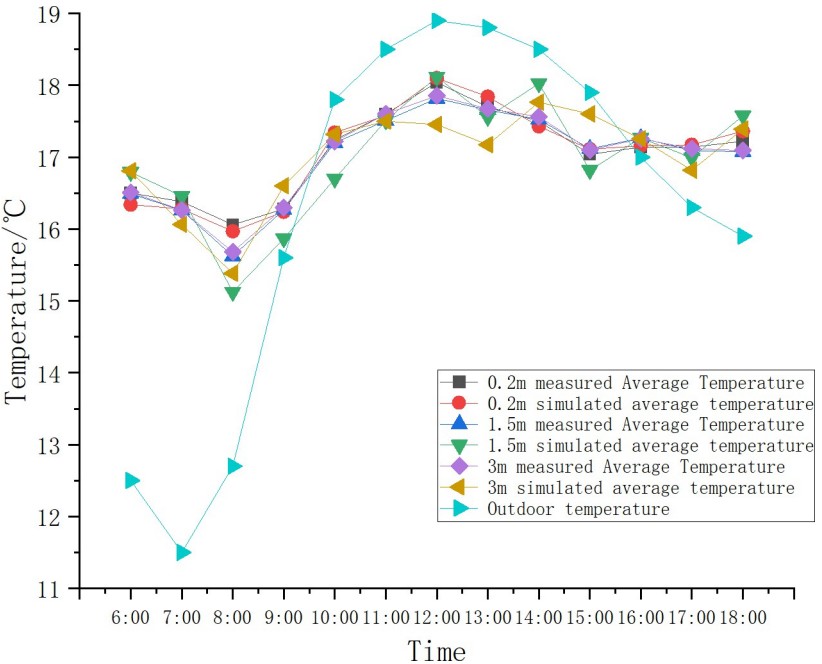

**Fig 10. Indoor temperature simulated and measured value curves.**

**Table 3. Hourly variation coefficients of outdoor temperature from 1 to 12 o'clock.**

| Time | 1:00 | 2:00 | 3:00 | 4:00 | 5:00 | 6:00 | 7:00 | 8:00 | 9:00 | 10:00 | 11:00 | 12:00 |
|------|------|------|------|------|------|------|------|------|------|-------|-------|-------|
| β | -0.35 | -0.38 | -0.42 | -0.45 | -0.47 | -0.41 | -0.28 | -0.12 | 0.03 | 0.16 | 0.29 | 0.4 |

## 4.2 Analysis of air temperature field distribution inside the workshop

As shown in Fig 11, the indoor temperature distribution on the east-west cross-section of the workshop is relatively uniform at 12:00, with temperatures ranging from 22 to 23˚C. The west side is the equipment room, where the temperature is slightly higher, between 23 and 24˚C. As shown in Fig 12, the indoor temperature distribution on the north-south cross-section is also relatively uniform at the same time. Since the water temperature is constant at 17˚C and the building wall temperature is 29˚C. As shown in Fig 13, the image of the velocity field along the Y-axis of the workshop indicates slow air movement near the walls, caused by natural convection due to temperature differences. This convection leads to convective heat transfer within the workshop's air, resulting in a reduced vertical temperature gradient.

Through simulation and prediction, the air temperature involved in heat exchange with the water body during normal operation of the workshop was determined. This outcome can be used in the initial stages of equipment selection for the workshop, to calculate the heat exchange results after the installation of equipment. Hence, it is evident that CFD plays an irreplaceable role in the selection of equipment. Through the results of the simulation, it is known that the overall temperature field in the breeding workshop is relatively uniform, so the average temperature at a cross-section 0.2m above the water surface is taken as the temperature for heat exchange with the water body, and the cooling load of the water body in the workshop is solved. The hourly average temperature of the air exchanging heat with the water body is shown in Tables 5 and 6.:

## 4.3. Calculation of cooling load of the water body

The cooling load of the water body primarily consists of three components: the heat transfer from the air to the water body, the heat transfer from the pool wall to the water body, and the heat transfer from the pool bottom to the water body. The calculation formula is as follows [25]:

$$Q_{\text{total}} = a_w(t_h - t_w)F_1 + F_2 K(t_h - t_w) + a_w(t_l - t_w)F_1 \tag{8}$$

$Q_{\text{total}}$-Summer water body heat exchange cooling load, kW;

$a_w$-Heat transfer coefficient of the water body, W/(m$^2$K). In this article, it is taken as 15 W/(m$^2$K);

$t_w$-Water body temperature,˚C;

$t_h$-Air temperature exchanging heat with the water body,˚C;

$t_l$-Bottom temperature exchanging heat with the water body,˚C;

$F_1$-Water pool surface area, m$^2$;

$F_2$-Water wall surface area, m$^2$;

**Table 4. Hourly variation coefficients of outdoor temperature from 13 to 24 o'clock.**

| Time | 13:00 | 14:00 | 15:00 | 16:00 | 17:00 | 18:00 | 19:00 | 20:00 | 21:00 | 22:00 | 23:00 | 24:00 |
|------|-------|-------|-------|-------|-------|-------|-------|-------|-------|-------|-------|-------|
| β | 0.48 | 0.52 | 0.51 | 0.43 | 0.39 | 0.28 | 0.14 | 0 | -0.1 | -0.17 | -0.23 | -0.28 |

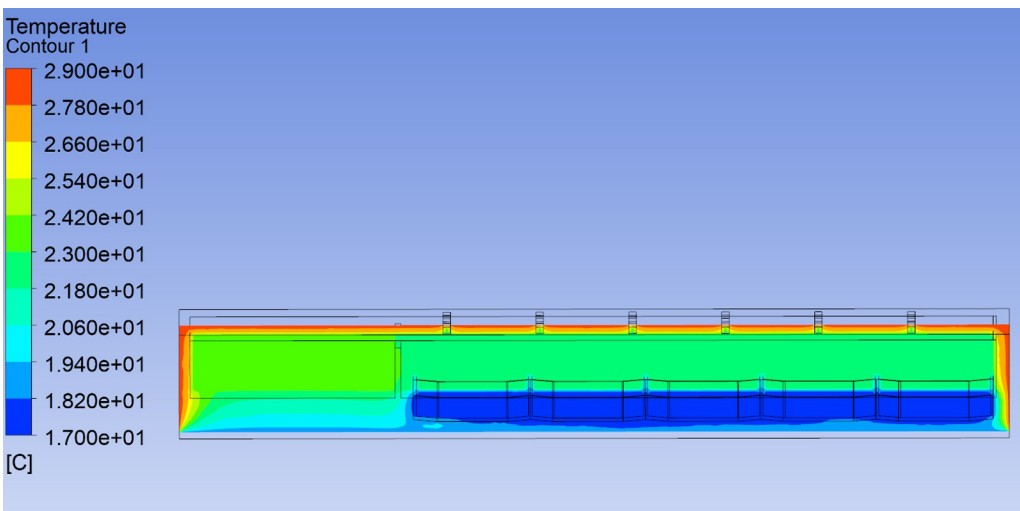

**Fig 11. The temperature distribution of section X-axis of the workshop at 12:00.**

K- Heat transfer coefficient, W/(m²K).

$$K = \frac{1}{\frac{1}{h_{water}} + \frac{\delta}{\lambda} + \frac{1}{h_{air}}} \tag{9}$$

$h_{water}$-Convective heat transfer coefficient of water, W/(m²K);
$h_{air}$-Convective heat transfer coefficient of air, W/(m²K);
$\delta$-Thickness of the pool, m;
$\lambda$-Thermal conductivity of the pool, W/(mK).

The contact area between the water body and the air in the workshop is 52.84m², the farming water body temperature is kept constant at 17°C, the contact area between the water body and the pool wall in the workshop is 43.28m². The convective heat transfer coefficient between the air and the pool wall in the workshop is 15W/(m²K), and between the farming water body

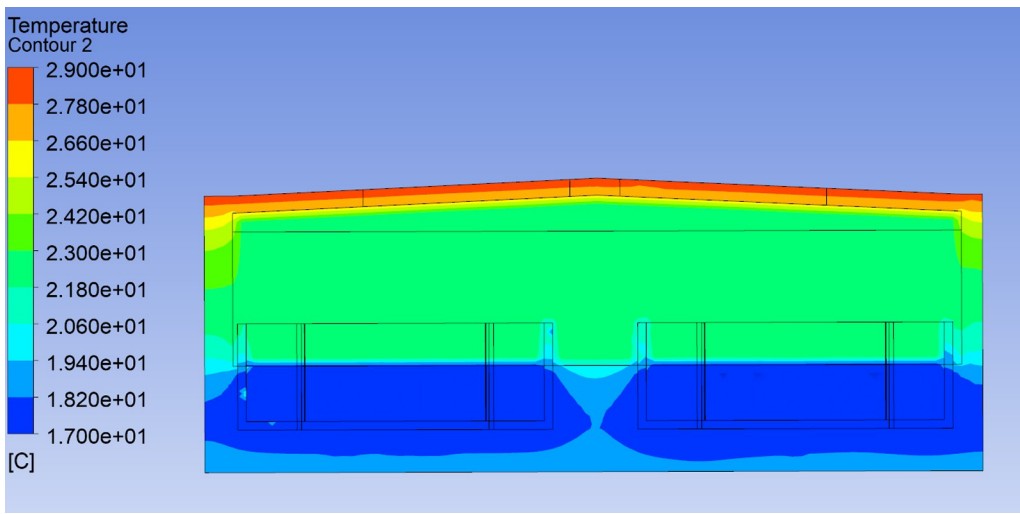

**Fig 12. The temperature distribution of section Y-axis of the workshop at 12:00.**

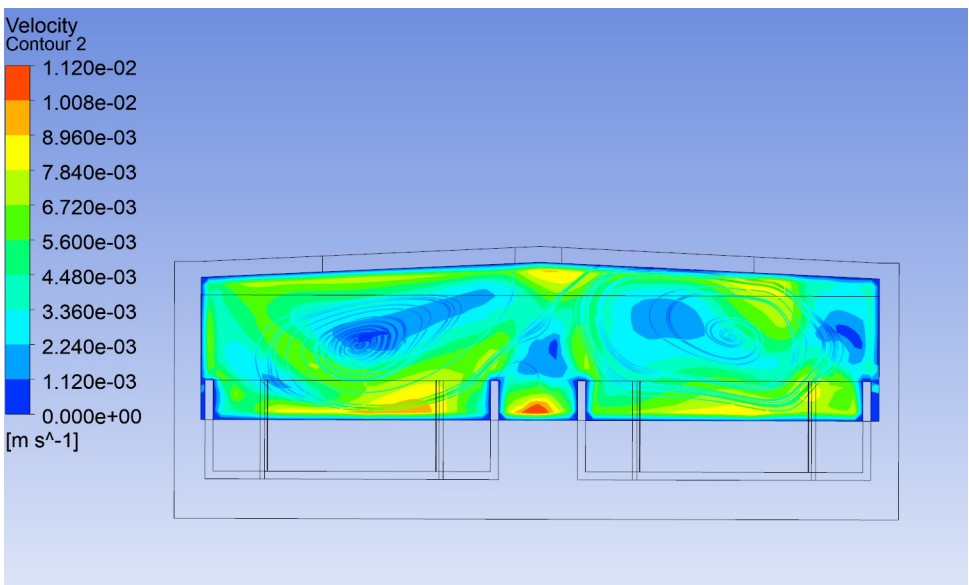

**Fig 13. The streamline field of section Y-axis of the workshop at 12:00.**

and the pool wall in the workshop it is 1000W/(m$^2$K). The thickness of the pool wall is 0.24m, and the heat transfer coefficient of the pool wall is 0.24 W/(mK). In summer, the temperature 1m underground in Dalian is 20°C. The hourly cooling load of the farming water body in the workshop is shown in Tables 7 and 8, as follows:

## 4.3. Selection of heat pump equipment

Considering the actual situation of the No. 2 workshop of Dalian Fuguyu Pufferfish Aquaculture Full Industry Chain, there are a total of 10 breeding pools in the workshop. At 4 pm in summer, the cooling load of the water body reaches the peak, which is 9.46KW, and the total cooling load of the workshop is 94.6KW. When designing the heat pump device, it is necessary to ensure that the cooling capacity of the heat pump can meet the maximum load of the fish pond. Based on the above calculation results, this study suggests the parameters of the water source heat pump unit as shown in Table 9.

## 5. Conclusion

In this study, a 3D unsteady-state CFD model was established for the industrialized recirculating aquaculture workshop, considering outdoor temperature and solar radiation changes over time, as well as phase change heat transfer of the water body. The root mean square error between the model simulation results and the experimental data was 0.45°C. The agreement between the experimental results and the simulation results were very good, proving the established CFD model to be effective. It can accurately describe the temporal variation and spatial distribution characteristics of the air temperature in the workshop.

**Table 5. Average air temperature above the water surface in the workshop from 1 to 12 o'clock in a typical summer climate.**

| Time | 1: 00 | 2: 00 | 3: 00 | 4: 00 | 5: 00 | 6: 00 | 7: 00 | 8: 00 | 9: 00 | 10: 00 | 11: 00 | 12: 00 |
|---|---|---|---|---|---|---|---|---|---|---|---|---|
| Temperature(°C) | 20.7 | 20.6 | 20.5 | 20.3 | 20.2 | 20.1 | 20.1 | 20.7 | 21.3 | 21.6 | 22.1 | 22.5 |

**Table 6. Average air temperature above the water surface in the workshop from 13 to 24 o'clock in a typical summer climate.**

| Time | 13:00 | 14:00 | 15:00 | 16:00 | 17:00 | 18:00 | 19:00 | 20:00 | 21:00 | 22:00 | 23:00 | 24:00 |
|---|---|---|---|---|---|---|---|---|---|---|---|---|
| Temperature(°C) | 22.9 | 23.2 | 23.5 | 23.6 | 23.3 | 22.9 | 22.3 | 21.9 | 21.5 | 21.4 | 21.1 | 20.8 |

**Table 7. Hourly cooling loads from water surface heat transfer from 1 to 12 o'clock.**

| Time | 1:00 | 2:00 | 3:00 | 4:00 | 5:00 | 6:00 | 7:00 | 8:00 | 9:00 | 10:00 | 11:00 | 12:00 |
|---|---|---|---|---|---|---|---|---|---|---|---|---|
| $t_h$(°C) | 20.7 | 20.6 | 20.5 | 20.3 | 20.2 | 20.1 | 20.1 | 20.7 | 21.3 | 21.6 | 22.1 | 22.5 |
| $Q_{total}$ (kW) | 7.04 | 6.94 | 6.94 | 6.73 | 6.62 | 6.52 | 6.52 | 7.04 | 7.57 | 7.78 | 8.2 | 8.62 |

**Table 8. Hourly cooling loads from water surface heat transfer from 13 to 24 o'clock.**

| Time | 13:00 | 14:00 | 15:00 | 16:00 | 17:00 | 18:00 | 19:00 | 20:00 | 21:00 | 22:00 | 23:00 | 24:00 |
|---|---|---|---|---|---|---|---|---|---|---|---|---|
| $t_h$ (°C) | 22.9 | 23.2 | 23.5 | 23.6 | 23.3 | 22.9 | 22.3 | 21.9 | 21.5 | 21.4 | 21.1 | 20.8 |
| $Q_{total}$ (kW) | 8.94 | 9.15 | 9.36 | 9.46 | 9.25 | 8.94 | 8.41 | 8.09 | 7.78 | 7.67 | 7.36 | 7.15 |

**Table 9. Parameters of the water source heat pump unit.**

| Model | Daikin GCHP-40MAH Ground Source Heat Pump |
|---|---|
| Rated Cooling Capacity | 40 kW |
| Rated Heating Capacity | 45.2 kW |
| COP | 4.09 |
| Compressor Type | Fixed-frequency Scroll Compressor |
| Quantity | 3 units |
| Size (Length × Width × Height) | 1220mm × 700mm × 1650mm |

This study used a pufferfish breeding workshop in Dalian as a case study, and simulated the indoor air temperature field of this workshop under typical summer outdoor temperature conditions. There is natural convection in the workshop, and the temperature is quite evenly distributed. The study took the average air temperature near the water surface to calculate the cooling load. The results showed that the peak cooling load of the workshop in summer occurred at 16:00, totaling 94.6KW. Based on the calculation results, this study suggests that the workshop use the Daikin GCHP-40MAH type ground source heat pump as the water temperature control equipment.

In this study, the CFD method was used to simulate the indoor air temperature field after the installation of the water temperature control equipment, and then the water temperature control equipment was selected based on the simulation results. This selection method based on simulation results scientifically solves the problem of precise calculation of equipment selection parameters, and provides a reference for the selection of water temperature control equipment in other aquaculture workshops.

## Supporting information

**S1 Dataset.**
(XLSX)

**S1 File.**
(DOCX)

**S1 Table.**
(XLSX)

## Author Contributions

**Conceptualization:** Xu Ziyun, Du Ping, Wan Jiacheng, Lin Leyao, Chen Kairui.

**Data curation:** Du Ping, Wan Jiacheng, Lin Leyao, Chen Kairui.

**Formal analysis:** Xu Ziyun, Wan Jiacheng, Lin Leyao, Chen Kairui.

**Investigation:** Xu Ziyun, Du Ping, Wan Jiacheng, Lin Leyao.

**Methodology:** Xu Ziyun, Du Ping, Wan Jiacheng, Lin Leyao, Chen Kairui.

**Project administration:** Wan Jiacheng, Chen Kairui.

**Resources:** Xu Ziyun, Wan Jiacheng, Lin Leyao, Chen Kairui.

**Software:** Lin Leyao.

**Supervision:** Xu Ziyun, Du Ping, Wan Jiacheng, Lin Leyao, Chen Kairui.

**Validation:** Du Ping, Wan Jiacheng, Lin Leyao, Chen Kairui.

**Visualization:** Chen Kairui.

**Writing – original draft:** Xu Ziyun, Du Ping, Wan Jiacheng, Lin Leyao, Chen Kairui.

**Writing – review & editing:** Xu Ziyun, Du Ping, Wan Jiacheng, Lin Leyao, Chen Kairui.

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
