## [Decision Letter · Decision Letter 0]

2 Jan 2024

PONE-D-23-39332Selection of Heat Pump Units for Recirculating Aquaculture Workshops Using Computational Fluid Dynamics: Numerical SimulationPLOS ONE

Dear Dr. Ziyun,

Thank you for submitting your manuscript to PLOS ONE. After careful consideration, we feel that it has merit but does not fully meet PLOS ONE’s publication criteria as it currently stands. Therefore, we invite you to submit a revised version of the manuscript that addresses the points raised during the review process.

**ACADEMIC EDITOR:**
1. Article lacks recent literature. Please add literature between 2020 to 2023.

2. More simulations required to justify CFD analysis.==============================

We look forward to receiving your revised manuscript.

Kind regards,

Goutam Saha, PhD

Academic Editor

PLOS ONE

Journal Requirements:

2.PLOS requires an ORCID iD for the corresponding author in Editorial Manager on papers submitted after December 6th, 2016. Please ensure that you have an ORCID iD and that it is validated in Editorial Manager. To do this, go to ‘Update my Information’ (in the upper left-hand corner of the main menu), and click on the Fetch/Validate link next to the ORCID field. This will take you to the ORCID site and allow you to create a new iD or authenticate a pre-existing iD in Editorial Manager. Please see the following video for instructions on linking an ORCID iD to your Editorial Manager account: https://www.youtube.com/watch?v=_xcclfuvtxQ

4. We suggest you thoroughly copyedit your manuscript for language usage, spelling, and grammar. If you do not know anyone who can help you do this, you may wish to consider employing a professional scientific editing service.  

Reviewers' comments:

Reviewer's Responses to Questions

**Comments to the Author**

1. Is the manuscript technically sound, and do the data support the conclusions?

Reviewer #1: Yes

Reviewer #2: Partly

Reviewer #3: Yes

Reviewer #4: Partly

2. Has the statistical analysis been performed appropriately and rigorously? 

Reviewer #1: N/A

Reviewer #2: N/A

Reviewer #3: Yes

Reviewer #4: No

3. Have the authors made all data underlying the findings in their manuscript fully available?

Reviewer #1: No

Reviewer #2: No

Reviewer #3: Yes

Reviewer #4: Yes

4. Is the manuscript presented in an intelligible fashion and written in standard English?

Reviewer #1: Yes

Reviewer #2: No

Reviewer #3: Yes

Reviewer #4: No

5. Review Comments to the Author

Reviewer #1: The following comments must be taken into consideration before publishing:

1. In the abstract, the authors must avoid general description and focusing on the methodology, the results, and conclusions.

2. In section 1. Introduction, the authors spent more than on page talking about the well-known information about CFD role for solving heat transfer model. This introduction must be reduced and the introduction should focus on the main paper objective. And more references must be added about the objective of the paper. Moreover, nothing mentioned about the CFD software that has been used by the authors.

3. In section 2.1, a schematic diagram for the physical model must be added to demonstrate the details that were mentioned in this section.

4. In section 2.2 Computational Domain and Grid Division, lines 175-179, the choosing of scheme 2 must be justified by drawing a graph between the values of highest temperature of the indoor air and the number of the Grids for each scheme.

5. It preferable to present a cross-section for the Grid mesh of Fig.1.

6. Add reference(s) for the equations 1-2.

7. In section 2.3 Basic Control Equations, lines-206-207, how did the authors decide that the flow is laminar? Justify the assumption by calculating Rayleigh number.

8. The range of application of Equation 3.

9. In section 2.5 Computational Domain and Boundary Conditions, The author must add the mathematical formulation of the boundary conditions mentioned in this section.

10. The details of the equation 5. Where did the authors get this equation?

11. In section 4.2, The Figure 5 should be Figure 6.

12. In section 4.2, a streamlines of velocity distribution.

13. In section 5, list the conclusions as bullet points.

Reviewer #2: The selection of water temperature regulation equipment plays a crucial role in the design of workshops. At present, the choice of water temperature control equipment is usually based on the volume of the fish pond and thermal parameters calculation, combined with aquaculture experience. Due to limitations such as the environment, climate, and types of fish, empirical formulas are only applicable under specific conditions, lacking universal reference value. With the popularization of computational fluid dynamics (CFD) calculation methods, this paper proposes to apply CFD simulation of the temperature field to accurately calculate the heat exchange value between indoor air and water, providing a new approach for equipment selection. I have made some comments as follows:

Major issues:

1. The novelty of this study needs to be clarified. Please describe this adequately.

2. The validation part needs to be clarified. In Figure 2, it is unclear how to get the experimental data.

3. The description of the experimental part is not elaborately described.

4. The section and subsection fronts need to be distinguished.

5. Provide proper reference for formulas 1 and 2.

6. Describe elaborately how the simulation results help control water temperature equipment. They need to be more apparent to the audience.

7. Where is the phase change mode working, and what is the interfacial condition for the phase change?

8. What is the flow governing parameters? If it is Reynolds number, then describe how to influence the Re of the flow and temperature pattern.

9. Is it force convection or mixed convection? Describe clearly.

10. The description of the CFD part needs to be better. The CFD study needs to be clarified here. The authors should elaborate on this and correlate the necessity of this CFD study.

Minor Issues:

Page 1

Abstract

Line 15: : “parameters” should be corrected as “paramerer”.

Line 25: “There should be a a before 12th word “reference”.

Page 1-3

1. Introduction

Line 39: “There should be a space after 4th word “methods”.

Line 41:“There should be replace and with comma.No need to repetation of and after 6th word “motion” we can write ''numerical methods to simulate fluid motion, heat, and mass transfer processes.".

Line 42: “in” should be corrected as “for”.

Line 45: “There should be a and after 11th word “temperature”.

Line 46: “There should be a The after 9th word “improving”.

Line 73: “There should be a The before 9th word “CFD”.

Line 73: “There should be a The after 8th word '''and'.

Line 118: “material” should be corrected as “materials”.

Page 4-9

2. Establishment and Solution of the CFD Model for Factory-Based Recirculating Aquaculture Workshops

Line 159: “There should no be Comma after 10th word "indoors".

Line 221: “There no need to repeat the word considers 7th word ''also".

Line 268: “are” should be corrected as “is” after the 5th word.

Line 268: “There should be a a after 8th word "as".

Line 268: “There should be a Hyphen between ''third" and "type" after 8th word "as". .

Line 271: “There should be a Hyphen between ''first" and "type" after 10th word "as".

Line 272: “bottom” should be corrected as “bottoms” after the 5th word.

Line 288: “There should be a The before 9th word “Zhuanghe”.

Line 298: “There should be Of The before 6th word “water”.

Page – 10

3. Experimental Design and Model Validation for a Factoryy-based Recirculating Aquaculture Workshop

Line 354: “There should be The before 3rd word “summer”.

Page – 11-16

4. Equipment Selection for Recirculating Aquaculture Workshop

Line 379: “There should be for after 5th word “allows”.

Line 458: “There should no be Comma after 3rd word "unit".

Line 453: “There should be the after 6th word “in”.

Page- 17

5. Conclusion

Line 472: “was” should be corrected as “were” after the 4th word "results".

Reviewer #3: I would recommend this manuscript as a candidate for publication in PLOS ONE Journal with major revisions. The authors might need to consider the following comments:

1. The title would be better if the author rewrite Title. The word CFD methods and A numerical simulation both are similar.

2. In Abstract, the problem statement is articulated well, identifying the specific issue or gap that the research aims to address. It would be beneficial to clearly define the numerical methods for solving the computational domain. It is recommended to provide a bit more detail on the key results or outcomes, ensuring that readers get a glimpse of the research's contribution.

3. The researcher has effectively outlined the objective and scope of the study. However, the review of existing literature appears to be insufficient, and there is a notable gap in addressing the current state of knowledge in the field of fluid-solid

coupling simulation method to describe the energy transfer process analysis. The author should expand the literature review section to encompass a more comprehensive overview of recent relevant studies, theories, and methodologies related to the research topic.

4. In section 2.1 detail of Physical Model of the Workshop Space, page 4, the description of designing workshop space with boundary condition is not clear. It would be very helpful for the readers if the author provides a Figure of physical model with detailed dimensions and boundary conditions of workshop space.

5. In section Meshing and validation, page 7, Did the authors use the block mesh and prism layer mesh for the refinement of the investigated area inside the workshop? As we know, the prism layer mesh is used with a core volume mesh to generate orthogonal prismatic cells next to wall boundaries. This layer of the cell is necessary to improve the accuracy of the flow solution. Moreover, I have not found any information regarding Grid independence test for mesh validation in this manuscript regarding this issue.

6. It mentions the use of the Finite Volume Method (FVM) but does not describe the methodology or approach in any detail. The mathematical treatment for unsteady simulation part is also not lucid. Please check and modify whatever is required.

7. In Section 2.3, the authors defined the state of indoor air can be regarded as three-dimensional, unsteady-state, incompressible Newtonian fluid motion, and equation 1 are not consistent with the mentioned references [10-14 and 15-17], like notations of subscripts and Boussinesq assumption are not appropriately used. Please check and correct me if I am wrong. Moreover, all vector natation’s and variables declarations are not adequate, which, lowers the scientific quality of the paper. Please check and modify whatever is required.

8. Figures provides valuable insights into the paper. However, a comprehensive evaluation suggests that there are areas to improve from Figure 2 to Figure 5. The title of Figures must be consistent with the results and level of Figures heading. Please correct all Figures. For example, “Fig. 4 Error Distribution Diagram between Simulated and Measured Values” is not consistent with the level of Fig 4.

9. It would be appreciated if the authors include the validation of the numerical results with published results.

10. There are many typographic mistakes in the text and those should be corrected.

11. The formatting in this manuscript and format of all references in this manuscript does not meet the requirement of this journal. Please see and follow the guideline for writing the manuscript of this journal.

Reviewer #4: 1. Writing of introduction should be improved as the sequence of keywords are not followed for potential readers.

2. References are not called properly and references [1-5] are cited many times in many places which can be handled properly.

3. Basic control equations requires more details specially how the radiation affects the flow and in which condition this should be addressed.

4. The numerical uncertainty due to the weather condition, i.e. sunny or cloudy, needs to be addressed. Only one day data for producing the results are not convincing.

5. Calculation method has not written with adequate information. The Blackbox of the commercial software FLUENT has not opened and discussed in the manuscript.

6. The given simulation results are not convincing to justify the need of CFD study for this type of workshop. Temperature variation are within 2-3 degree Celsius. Placing thermal sensors for few places are adequate to read temperature data for several days or so. Thereafter, Machine Learning Approach (MLA) can be utilized to predict the time of maximum cooling load. What the significance of study CFD that supersede the modern MLA approach.

6. PLOS authors have the option to publish the peer review history of their article (what does this mean?). If published, this will include your full peer review and any attached files.

Reviewer #1: **Yes: **Ahmed Abdul-Rida Al-Waaly

Reviewer #2: No

Reviewer #3: No

Reviewer #4: **Yes: **Dr. Preetom Nag

---

## [Decision Letter · Decision Letter 1]

20 Feb 2024

Optimizing Heat Pump Unit Selection for Recirculating Aquaculture Workshops through Computational Fluid Dynamics

PONE-D-23-39332R1

Dear Dr. Ziyun,

We’re pleased to inform you that your manuscript has been judged scientifically suitable for publication and will be formally accepted for publication once it meets all outstanding technical requirements.

Kind regards,

Goutam Saha, PhD

Academic Editor

PLOS ONE

Additional Editor Comments (optional):

Accepted as it is.

Reviewers' comments:

Reviewer's Responses to Questions

**Comments to the Author**

1. If the authors have adequately addressed your comments raised in a previous round of review and you feel that this manuscript is now acceptable for publication, you may indicate that here to bypass the “Comments to the Author” section, enter your conflict of interest statement in the “Confidential to Editor” section, and submit your "Accept" recommendation.

Reviewer #1: All comments have been addressed

Reviewer #2: All comments have been addressed

Reviewer #3: All comments have been addressed

Reviewer #4: All comments have been addressed

2. Is the manuscript technically sound, and do the data support the conclusions?

Reviewer #1: Yes

Reviewer #2: Yes

Reviewer #3: Yes

Reviewer #4: Yes

3. Has the statistical analysis been performed appropriately and rigorously? 

Reviewer #1: N/A

Reviewer #2: N/A

Reviewer #3: Yes

Reviewer #4: Yes

4. Have the authors made all data underlying the findings in their manuscript fully available?

Reviewer #1: Yes

Reviewer #2: No

Reviewer #3: Yes

Reviewer #4: Yes

5. Is the manuscript presented in an intelligible fashion and written in standard English?

Reviewer #1: Yes

Reviewer #2: Yes

Reviewer #3: Yes

Reviewer #4: Yes

6. Review Comments to the Author

Reviewer #1: (No Response)

Reviewer #2: Recently, I have reviewed this manuscript and made some major comments. The authors have revised the manuscript based on my comments.

Reviewer #3: The selection of water temperature regulation equipment plays a crucial role in the design of workshops. At present, the choice of water temperature control equipment is usually based on the volume of the fish pond and thermal parameters calculation, combined with aquaculture experience. Due to limitations such as the environment, climate, and types of fish, empirical formulas are only applicable under specific conditions, lacking universal reference value. With the popularization of computational fluid dynamics (CFD) calculation methods, this paper proposes to apply CFD simulation of the temperature field to accurately calculate the heat exchange value between indoor air and water, providing a new approach for equipment selection. In summary, happily, I would recommend this paper as a candidate for publication in PLOS ONE Journal.

Reviewer #4: (No Response)

7. PLOS authors have the option to publish the peer review history of their article (what does this mean?). If published, this will include your full peer review and any attached files.

Reviewer #1: No

Reviewer #2: No

Reviewer #3: **Yes: **Dr. Bijan Krishna Saha

Reviewer #4: **Yes: **Preetom Nag, Ph.D.

---

## [Editor Report · Acceptance letter]

1 Apr 2024

PONE-D-23-39332R1 

PLOS ONE

Dear Dr. Ziyun, 

I'm pleased to inform you that your manuscript has been deemed suitable for publication in PLOS ONE. Congratulations! Your manuscript is now being handed over to our production team.

Kind regards, 

on behalf of

Dr. Goutam Saha 

Academic Editor

PLOS ONE